# Feeding Flaxseed and Lupins during the Transition Period in Dairy Cows: Effects on Production Performance, Fertility and Biochemical Blood Indices

**DOI:** 10.3390/ani13121972

**Published:** 2023-06-13

**Authors:** Ioannis Nanas, Stella Dokou, Labrini V. Athanasiou, Eleni Dovolou, Thomas M. Chouzouris, Stelios Vasilopoulos, Katerina Grigoriadou, Ilias Giannenas, Georgios S. Amiridis

**Affiliations:** 1Department of Obstetrics and Reproduction, Veterinary Faculty, University of Thessaly, 43100 Karditsa, Greece; 2Laboratory of Nutrition, Faculty of Veterinary Medicine, School of Health Science, Aristotle University of Thessaloniki, 54124 Thessaloniki, Greece; 3Department of Medicine, Veterinary Faculty, University of Thessaly, 43100 Karditsa, Greece; 4Laboratory of Reproduction, Department of Animal Science, University of Thessaly, 41223 Larissa, Greece; 5ELVIZ Hellenic Feedstuff Industry S.A., 59300 Plati, Greece; 6Institute of Plant Breeding and Genetic Resources, Hellenic Agricultural Organization–DEMETER, 57001 Thermi, Greece

**Keywords:** dairy cows, soybean meal, flaxseed, lupins, fertility, blood indices

## Abstract

**Simple Summary:**

The use of alternative proteinaceous feedstuffs is under investigation worldwide, due to the increased economic and ecological burdens of soybean meal, especially for dairy farms. Here, we examined whether the partial substitution of soymeal by a mixture of locally produced flaxseed and lupins could affect milk yield and fertility parameters of dairy cows during the transition period. Milk yield and composition were examined. The dietary treatment affected neither milk yield nor its composition. Animals offered the diet with flaxseed and lupin seeds expressed the first postpartum estrus and conceived earlier than control cows. These results imply that the replacement of soymeal with flaxseed and lupin is a profitable feed modification, which benefits overall fertility in dairy cows.

**Abstract:**

Flaxseed and lupin seed were offered as an alternative dietary approach in dairy cows, through the partial substitution of soybean meal. Milk production and fertility traits were investigated. A total of 330 animals were allocated into two groups, treated (*n* = 176) and control (*n* = 154). From each group, 30 animals were selected for hematological and cytological studies. The experimental feeding period lasted for 81 days (25 days prepartum and 56 days postpartum). The control ration (group C) contained corn, barley, soybean meal, rapeseed cake, corn silage and lucerne hay; whereas, in the treatment group (group T), 50% of the soybean meal was replaced by an equal mixture of flaxseed and lupins. The two rations were formulated to be isonitrogenous and isoenergetic. Milk samples were analyzed for chemical composition, somatic cell count (SCC) content and total colony forming units (CFU). Blood samples were collected, and serum was analyzed for non-esterified fatty acids (NEFA), acute phase proteins (haptoglobin and serum amyloid) and lipid oxidation indices, namely thiobarbituric-acid-reactive substances (TBARS) and catalase activity. To assess polymorphonuclear neutrophils (PMN) numbers, endometrial samples from each cow were collected on days 21 and 42. No difference was recorded between groups in milk yield (*p* > 0.05). In multiparous cows, NEFA (mMol/L) concentrations were significantly lower in group T than in group C on day 14 (*p* > 0.009) and on day 42 (*p* = 0.05), while no difference was detected in the group of primiparous cows. At all time points, serum TBARS and catalase values were similar in both groups (*p* > 0.05). Multiparous cows in group T expressed the first postpartum estrus and conceived earlier than cows in group C (*p* ≤ 0.05). Between days 21 to 42 postpartum, the PMN reduction rate was higher in group T animals (*p* ≤ 0.05). Acute phase protein levels were in general lower in group T animals, and at specific time points differed significantly from group C (*p* ≤ 0.05). It was concluded that the partial replacement of soybean meal by flaxseed and lupins had no negative effect on milk yield or milk composition, and improved cow fertility; which, along with the lower cost of flaxseed and lupins mixture, may increase milk production profitability.

## 1. Introduction

Soybean meal is traditionally used as the primary protein source in dairy cow rations, due to its high protein (496 g/kg of DM) and relatively high energy content (1.26 UFL/kg DM) [1], which are required for the support of milk synthesis. The concept of partial or total soybean meal substitution has been emerging in the European Union countries, based on the increased environmental burden of soybean cultivation and overseas transportation, along with the fluctuating soybean cost, which is controlled by international stock markets, leading to increased economic pressure on farmers [2]. This situation is further exacerbated in countries, such as Greece, where the local soybean production is almost non-existent [3], resulting in large-scale imports.

Semi-arid areas with low rainfall cannot support growth of grass to satisfy dietary needs of dairy cows. The need for alternative sources of cost-effective protein, suitable to decrease animal and overall farm environmental impact, is vital for the sustainability of the primary sector. This situation is further complicated due to climate change, which is manifested by extended drought periods and unusual floods, causing further degradation of the low productivity arable land. Hence, the selection of crops able to be grown in low productivity arable land and provide valuable feedstuffs appears as key challenge for the sustainability of the dairy sector in most of the Mediterranean countries [4]. 

Flaxseed (*Linum usitatissimum*) is an oil-rich seed with high protein content (226 g/kg of DM) and high energy density (1.69 UFL/kg DM), making it an attractive feedstuff for dairy cows [1,5]. Flaxseed’s nutritive value is further enriched by its lipid profile, characterized by the increased unsaturated fatty acid (UFA) content; omega-3 fatty acids (ω-3) account for more than 50% of the total lipids. These fatty acids (FAs) are positively associated with human health by supporting cardiovascular function [6]. Concurrently, ω-3 were investigated as cow fertility enhancers and proven to support estrus length and intensity of cow’s typical estrus behavior [7]. Although rumen microbial metabolic activities cause UFA biohydrogenation, a process which modifies UFA to saturated fatty acids, it was shown that the inclusion of 25 g/kg DM of flaxseed oil in dairy cow rations resulted in increased ω-3 fatty acid concentrations in milk [8]. Moreover, locally produced flaxseed can be cost-effective compared to imported soybean meal in terms of energy, protein and fiber values. Additionally, another legume with considerably lower cost compared to soybean meal is white lupin, that, in combination with flaxseed, may replace soybean meal. A further increase on the cultivation of both seeds can result in reduction of the unit production cost, due to economical scale effects on seed production, fertilization, plant protection and mechanical harvesting. 

White lupin (*Lupinus albus*) is a legume of which the seeds also contain high protein and energy content (380 g/kg DM, 1.43 UFL/kg DM) and a satisfactory fibrous content [1]. Lupins can be purchased at a considerably lower cost compared to commonly used protein feedstuffs, and they also have minimal requirements for storing and handling [9]. Another main characteristic of lupin seeds is their tolerance to abiotic stresses, allowing them to grow in poor, acidic and/or contaminated soils [10,11]. Due to the combination of these key points, lupin seeds can be regarded as a suitable feedstuff for ruminants. However, lupins are low in sulfur-containing amino acids, such as methionine, and their proteins are characterized by rapid degradation in the rumen environment, which could present limitations in high-yield dairy cow rations [12]. Documented results on the possible effects of lupin seeds on the reproductive function of dairy cows are scarce. 

The transition period of dairy cows (three weeks before to three weeks after parturition) is associated with a rapid increase in nutrient requirements, along with a significant decrease in voluntary dry matter intake (DMI), thus creating a stressful period for the cow [13]. This situation predisposes the periparturient cow to a negative energy balance (NEB) status [14]. Metabolic disorders, reduced milk yield and pregnancy rate, along with reproductive disorders, such as retained placenta, metritis, subclinical endometritis and delayed conception, are associated with NEB during the transition period [15]. These metabolic and reproductive conditions may affect the cow’s productivity and future fertility during the upcoming lactation [16]. 

This study investigated the effects of partial soybean meal substitution with a mixture of flaxseed and white lupin seed (FL) in dairy cows’ diet. The hypothesis to be tested was that feeding them this mixture would sustain milk production and support general health and fertility in Holstein cows.

## 2. Materials and Methods

### 2.1. Ethics Guidelines of the Animal Research

All animals received humane care. The procedures described herein cohered to the suggestions of EU regulation (np. 2010/63/EU) [17] for the protection of animals used for scientific purposes, and were approved by the Ethical Committee of Animal Welfare of the University of Thessaly (license number 112/19/9/20).

### 2.2. Animals and Experimetal Design

The field study was carried out in a commercial dairy farm in central Greece with 530 milking Holstein cows, with an average milk production of 11.367 L/cow per lactation. All animals were fed a total mixed ration (TMR) twice per day, fresh drinking water was available ad libitum and cows were milked thrice daily at 5:30, 12:30 and 19:30, in a 24-parallel milking parlor with individual ear-tag identification. Feed samples were collected weakly and analyzed for their chemical composition using NIR spectrometer (PerkinElmer DA 7250, Perten INSTRUMENTS, Hägersten, Sweden).

One month prior to the commencement of the experimental phase, the animals were selected from the existing groups of dry cows and pregnant heifers, based on mean parity and BCS, by ear-tag recognition. The selected animals had a typical dry period (60 days prior to the expected calving) and then they were randomly allocated to one of the two treatment groups. In total, 330 animals were used, and the study was carried out during the thermoneutral period of the year (December to May).

The control group (C) consisted of multiparous cows (group CC, *n* = 104) with mean parity 3.2 ± 1.6 and BCS 3.6 ± 0.4 and heifers (group CH, *n* = 50) with BCS 3.7 ± 0.3, while the treated groups (T) consisted of multiparous cows (group TC, *n* = 124) with mean parity 3.4 ± 1.6 and BCS 3.6 ± 0.3 and heifers (group TH, *n* = 52) with BCS 3.7 ± 0.3). From each group, 30 animals (15 cows and 15 heifers) were randomly selected to record daily dry matter intake. Feed residuals, calculated as the difference of the weight of feed offered minus the weight of the feed left in the trough prior to the next feeding, were taken under consideration for the evaluation of dry matter intake. A subgroup of these animals was also used for the hematological and cytological inspection.

Animals in group C were offered a typical dairy cow diet (TMR) based on forages, cereal grains and soybean meal, following INRA recommendations [1]. The diet offered to group T animals was modified by substituting half of the soybean meal with an equal quantity of a mixture of flaxseed and lupin seed. Further adjustments were made for the quantity of corn, barley, rapeseed meal and molasses to formulate isocaloric and isonitrogenous rations (Table 1). The lupin seeds used in the experiment were of the drought resistant “Multitalia” variety of *Lupinus albus*, with a reduced alkaloid content and with a crude protein (CP) content of 35%. The flaxseed belonged to the *Linum usitatissimum* variety “Galaad”, with a CP content of 25%. Both crop species were cultivated in the area of Imathia, Macedonia, Northern Greece for this purpose. The control and modified TMR were offered to the corresponding groups from 25 days prepartum until day 56 postpartum.

### 2.3. Physical Evaluations and Clinical Examinations

Body condition score (BCS), dry matter intake, and clinical examinations, including assessment of uterine involution, ovarian resumption (verified by the identification of a corpus luteum) and pregnancy diagnosis were performed on a weekly basis by a veterinary surgeon, who visited the farm twice per week. A five-point scale was used to evaluate the BCS, ranging from emaciation score 0 to obesity score 5 [18]. Data were recorded for each cow individually.

### 2.4. Determination of Milk Yield and Composition

Every day, milk yield was individually recorded by the milking parlor recording system. Individual milk samples were collected during the morning milking from each quarter, dispensed into a 50 mL tube with the addition of potassium dichromate. The samples were analyzed for total fat, total protein, lactose, total solids (TS) and colony-forming units (CFU) by near-infrared spectroscopy using a MilkoScan 4000 (FOSS Electric, Integrated Milk Testing™, Hilleroed, Denmark). Somatic cell counts (SCC) were determined using a Fossomatic 5000 Basic (FOSS Electric, Hilleroed, Denmark). 

### 2.5. Hematological and Biochemical Examinations

Blood samples from each animal were collected on days −7, 0, 7, 14, 21 and 42, and analyzed for non-esterified fatty acids (NEFA) and acute phase protein concentrations. The blood samples were withdrawn from the coccygeal vein into plain and EDTA containing tubes. All tubes were centrifugated to collect serum and plasma, respectively. Plasma and serum were stored at −20 °C until analyzed for NEFA, haptoglobin (Hpt) and serum amyloid (SAA). Plasma NEFA concentrations were measured using a commercial kit (NEFA FS, DiaSys Diagnostics Systems GmbH, Holzheim Germany) according to the manufacturer’s instructions in an automatic biochemical analyzer ADVIA 1200 (Siemens Healthcare Diagnostics Inc., Tarrytown, NY, USA). Acute phase proteins were determined in blood samples collected on days −7, 7 and 21 using commercially available ELISA Kits (MyBioSource, San Diego, CA, USA). The sensitivity of the Hpt assay was 0.02 ng mL^−1^, and intra-assay and inter-assay coefficients of variation for serum samples containing low and high concentrations were 5.1%, 5.7%, 4.5% and 7.7%, respectively. In the SAA assay, the sensitivity was 0.1 ng mL^−1^ and intra-assay and inter-assay coefficients of variation for serum samples containing low and high concentrations were 5.9%, 7.5%, 9.5% and 6.7%, respectively.

Serum progesterone concentration was determined by a commercial ELISA kit (NRG, Marburg, Germany). Sensitivity of the assay was 0.08 ng ml^−1^ and the intra-assay and inter-assay coefficients of variation for serum samples containing low and high concentrations were 5.4%, 6.8%, 9.7% and 5.6%, respectively.

Pregnancy-associated glycoprotein (PAG) concentration was determined in serum by a commercial ELISA (DG29 kit, Conception Animal, Beaumont, QC, Canada), based on the determination of a specific embryonic glycoprotein that constitutes strategic and confidential corporate information. The sensitivity and the specificity of the assay were 96.8% and 98.7%, respectively [19]. All assays were performed in duplicates, and mean values were calculated. If differences > 5% were detected between sample duplicates, such samples were re-assayed.

Thiobarbituric acid-reactive substances (TBARS) in serum samples were determined according to Ahn et al. [20] with minor modifications. Briefly, blood samples were collected on day 42 of the experiment, and were centrifugated immediately after collection to isolate the serum and 1 mL aliquots were homogenized in 5 mL of distilled water with Ultra-Turrax T25 (Janke & Kunkel, IKA Labortechnik, IKA^®^-Werke GmbH & Co. KG, Staufen, Germany) for 15 s. Supplementarily, 1 mL of serum was stored in the refrigerator in order to repeat the TBARS analysis 3 days after the collection. Then, 1 mL aliquots of the homogenates were transferred to test tubes, and 10 μL of butylated hydroxyanisol (7.2%) and 1 mL of TBA-trichloroacetic acid solution (20 mM TBA in 15% trichloroacetic acid) were added. The sample mixtures were vortex-mixed and incubated in boiling water for 15 min. Following cooling, the samples were centrifuged at 1000× *g* for 15 min and the absorbance of each supernatant was measured at 532 nm with a spectrophotometer (UV 1700 PharmaSpec, Shimadzu, Japan). Serum oxidation was determined by estimating the 2-thiobarbituric acid-reactive substance (TBARS) values, and expressed as nanograms of malondialdehyde per ml of serum.

The activity of catalase was determined by an assay of hydrogen peroxide based on the formation of a stable complex with ammonium molybdate [21]. In brief, 10 μL of blood serum, collected on day 42 of the trial, was incubated at room temperature for 4 min in a 1 mL reaction mixture of 65 mM of hydrogen peroxide in 60 mM of sodium phosphate buffer. To terminate the reaction, 1 mL of 32.4 mM ammonium molybdate was added and absorbance was measured at 405 nm in a spectrophotometer (Shimadzu, Model UV-160A, Tokyo, Japan). The recorded absorbance was compared to a reference curve with known concentrations of hydrogen peroxide. Results of catalase activity are expressed in kU/L.

### 2.6. Uterine Examinations

During the first week postpartum (pp), all cows were examined per vagina for the presence of abnormal uterine content and treated accordingly, if needed. Diagnosis of post-parturient metritis was confirmed in cases of enlarged uterus with a red-brown watery vaginal discharge and a rectal temperature of ≥39.5 °C; such cows were treated with 2.0 mg of ceftiofur hydrochloride per kg of BW (Excenel, Zoetis, Belgium) 3 times every 24 h and 2.2 mg/kg BW of Flunixin meglumine (Finixin, MSD, Germany), for 2 days. After day 21 pp, all animals were examined for clinical endometritis with Metricheck (Simcro, Hamilton, New Zealand). The vulva lips were disinfected, and individually packed sterilized 4 cm silicon hemispheres were used for each cow. The hemisphere was moisturized with physiological saline before insertion in the vagina. For the evaluation of the vaginal content, a scale of 0 to 4 was used, where: score 0 = clear, transparent mucus (healthy cows); 1 = clouded mucus in the absence of pus (mild endometritis); 2 = mucus containing focal pus flakes; 3 = discharge containing less than 50% white mucopurulent mucus; 4 = discharge composed of more than 50% pus and/or sanguineous material. All animals diagnosed with a vaginal discharge score of 2 or 3 were treated with two PGF_2α_ injections (0.5 μg cloprostenol, 2 mL Estrumate MSD, Germany) administered 11 to 14 days apart, while cows with a score of 4 received the same treatment along with an intrauterine infusion of 500 mg Cephapirine (Metricure 1 MSD, Germany) administered 5 to 7 days after the first PGF_2α_ treatment. All treated animals were re-examined 7 to 19 days after treatment completion and no further treatment was applied unless a specific condition prevailed.

### 2.7. Ovarian Resumption, Estrus Expression and Pregnancy Diagnosis

Data on estrus expression were collected by the farm manager, and by analyzing the data on the electronic monitoring system that was installed in the farm (SCR, Cow Monitoring Systems, Allflex, Israel). The day of ovarian resumption was defined as the day of the first identification of a corpus luteum at rectal palpation, and confirmed by determining the progesterone concentration on day 7 of the cycle.

Pregnancy diagnosis was performed on days 29 to 36 post-insemination, by determining the serum pregnancy associated glycoproteins (PAG) concentration, and it was confirmed 10 to 15 days later by rectal palpation.

### 2.8. Endometrial Cytology

At days 21 and 42 pp, endometrial samples from animals categorized to score 1 at Metricheck were collected using separate pre-sterilized instruments for cytological examination. Fourteen multiparous cows (group TC *n* = 8 and group CC *n* = 6), and fifteen primiparous cows (group TH *n* = 8 and group CH *n* = 7) were examined. A cytology brush (Andwin Scientific, Simi Valley, CA, USA) was attached to a sterile metal stem and then placed into a stainless steel tube for easy passage through the cervix. To prevent contamination from the vaginal environment, a sanitary plastic cover was placed outside the tube. Prior to insertion, the perineal area was cleaned with a warm water solution of povidone iodide and sanitized with ethanol solution. The stainless steel tube was introduced to the uterine body, where the cytology brush was gently advanced and two to three rotational movements of the metal stem were smoothly performed, and the brush was then retracted in the stainless steel tube. Slides were prepared immediately after the collection by rolling the brush onto clean, glass microscope slides (Hirschmann Laborgeräte GmbH & Co. KG, Eberstadt, Germany). The smears were air-dried, fixed with methanol to maintain cellular morphology, Giemsa stained and stored pending microscopy. Cytological evaluation was performed under a light microscope, first at 100× and then at 400× magnification. The slides were blinded, assessed twice by the same operator, and the average of the two assessments was used for statistical analysis. In total, 300 nucleated cells were counted per slide and individual cell types were identified, including endometrial epithelial cells and PMNs [22]. Finally, the percentage of polymorphonuclear cells was calculated as the number of PMNs divided by the sum of PMNs and epithelial cells [23].

### 2.9. Statistical Analysis

Statistical analyses were performed using IBM SPSS Statistics 25.0 for Windows. The results were expressed as means ± SD. Data were checked for deviation from normal distribution (Shapiro–Wilk normality test) and for violation of assumption of homogeneity of variance (Levene’s test) for each comparison. Student’s *t*-test and Mann–Whitney U test were performed for continuous data as appropriate. The incidence of diseases was tested with a ×2 test. A one-way repeated measures ANOVA was used to show differences between time points and groups. Sphericity was tested with Mauchly’s test. The cytology data were analyzed with a commercial statistical software package (JASP 16.0). The normality of the data was tested with a Shapiro–Wilk test and the homogeneity of the variances with Levene’s test. Repeated measures analysis of variance were used to assess the significance of the differences of PMNs between days 1 and 2 within each group of animals. Post hoc comparisons were done using the Tukey test. An independent samples *t*-test was also run to compare the alteration rate of PMNs among groups. All comparisons were made at a significance level of *p* ≤ 0.05.

## 3. Results

### 3.1. Nutrotional Values, Body Condition Scoring and Milk Yield and Composition

Dry matter intake was not affected by the alteration of diet composition. Both diets had similar crude protein content. The diet of group T had increased fat content (100 g/DM of feed) compared to group C (75.5 g/DM of feed), however, group T had lower starch and sugar continent compared to group C. 

At any time point, no difference (*p* > 0.05) was detected in BCS between groups (group C 3.9 ± 0.4, 3.8 ± 0.2, 3.3 ± 0.6, group T 3.8 ± 0.5, 3.6 ± 0.4, 3.1 ± 0.4 for days 0, 14 and 21, respectively). 

During the first 56 days of lactation, no differences were detected in the mean daily milk production between treated and control groups, neither in cows nor in heifers (*p* > 0.05) (Figure 1), nor in dry matter intake (Table 2). Similarly, no differences (*p* > 0.05) were recorded in the mean of the first 56 days’ milk yield, fat, protein, SCC and CFU (Table 3).

### 3.2. Incidence of Diseases and Ovarian Resumption, Days to Conception

During the first 120 days pp, no difference (*p* > 0.05) was recorded in the incidence of common diseases between groups. Almost half of the health disorders were mainly mild or subclinical endometritis and required no treatment. It is worth mentioning that, although the period of dietary treatment lasted for 56 days pp, the carry-over effects on fertility and health parameters had to be evaluated later in the pp period, well after the 56-day time window, and up to day 120 pp.

When all animals were examined together, the calving to first spontaneous pp estrus interval tended to be shorter in group T; no differences were detected in first calf heifers, while multiparous treated cows expressed earlier the first heat. Details on calving to first interval are given in Table 4. Progesterone concentration in blood samples collected during diestrus of the first pp cycle (days 7 to 9 and 12 to 13 of the estrous cycle) did not differ between groups (group CC 4.6 ± 0.9 ng/mL and 5.2 ± 1.1 ng/mL, group TC 4.1 ± 1.1 ng/mL and 5.0 ± 0.8 ng/mL; CH 4.8 ± 1.2 ng/mL and 5.0 ± 0.6 ng/mL, group TH 4.8 ± 0.8 ng/mL and 5.0 ± 1.3 ng/mL for days 7–9 and 12–13, respectively). At 120 days pp, no statistically significant (*p* = 0.09) difference was detected on the overall proportion of pregnant animals between the control and treated groups. For the same period, the pregnancy rate was increased for the treated multiparous cows compared to the control, while no difference was detected for the primiparous cows (Table 5). 

### 3.3. Serum TBARS and Catalase Activity

Serum TBARS were determined at days 1 and 3 after refrigerated storage, as lipid oxidation may be formed; the blood samples collection was similar for both groups (*p* > 0.05) (Table 5). Similarly, no significant differences were noted between the different diets in terms of catalase activity (*p* > 0.05) (Table 6).

### 3.4. Serum NEFA, Hpt and SAA Values

NEFA (mMol/L) concentration tended to differ significantly between time points (Days −7, 0, 7, 21 and 42) in the control group (*p* = 0.069), but not in the treated group (*p* = 0.186) (Table 7). There was no statistically significant effect by group in the pattern of NEFA (mmol/L) concentration (*p* = 0.364). Between groups, the NEFA levels were significantly lower in group T than in group C on day 14 (*p* > 0.009) and on day 42 (*p* = 0.05). No difference was detected in the group of primiparous cows (*p* > 0.05).

Τhe results of Hpt concentrations (μg/μL) from healthy control and treated animals are presented in Table 7. In both treatment groups, multiparous cows presented statistically different (*p* < 0.001) mean Hpt concentration values between time points. In the treatment group (TC), 7 days before calving, the mean Hpt concentration was lower (*p* = 0.007) compared to the control group (CC). Treatment had no significant effect on the pattern of Hpt concentration at different time points (*p* = 0.248). Regarding the primiparous cows, mean Hpt concentration in group CH differed (*p* < 0.001) between time points. Similarly, a repeated measures ANOVA (sphericity assumed) detected differences (*p* = 0.004) among time points in group TH. Compared to the control treatment (CH), the mean Hpt values of the treatment heifers (TH) were lower pre-partum (*p* < 0.001), as well as 7 days pp (*p* = 0.016). Finally, the pattern of Hpt concentration among time points was affected by treatment (*p* = 0.003).

The SAA concentration results for multiparous and primiparous cows are presented in Table 7. Regarding multiparous cows, the mean SAA concentration differed significantly between time points in group CC (*p* = 0.010), as well as in group TC (*p* = 0.006). No treatment effect was observed in the pattern of SAA concentration (*p* = 0.291)**.** Between groups, SAA levels differed on day −7 (*p* = 0.051) and tended to differ on day 21 (*p* = 0.08), whereas no difference was detected on day 7. The mean SAA concentration in primiparous control cows (CH) differed (*p* = 0.011) between time points, whereas no significant difference (*p* = 0.098) was detected among time points in the treated group (TH). Finally, the pattern of SAA concentration among time points was not affected by treatment group (*p* = 0.257). The mean SAA concentration was lower (*p* = 0.011) in pre-partum treated heifers (TH) than in the control heifers (CH). No differences between treatment groups were detected on days 7 and 21 (*p* = 0.461 and *p* = 0.174, respectively).

### 3.5. Cytology

In the control group of multiparous cows, the average PMNs (%) remained unaffected (*p* > 0.05) between day 21 and day 42 (Figure 2). In half of the cases, PNMs values were increased and the other halves were reduced; the average reduction was 14.41 ± 24.16%. In treatment group, average PMNs values were significantly lower on day 42 compared to day 21 (*p* < 0.05). The reduction was observed in all cases, and was at an average rate of 82.67 ± 4.6%, and was significantly higher than in control group (*p* < 0.05). In heifers, the average PMNs values of the treatment group were significantly lower on day 42 compared to day 21 (*p* < 0.05). Although the average PMNs in control group was also lower on day 42 than day 41, the difference was not significant (*p* > 0.05; Figure 3). The reduction was observed in all cases of both groups, and the reduction rates were not significantly different among groups (46.23 ± 10.74 and 25.42 ± 11.04 for treatment and control group, respectively).

## 4. Discussion

Diet cost represents the primary input expense at any livestock breeding operation. One strategy to reduce feed cost would be the substitution of soybean meal with alternative lower-cost, but effective, feedstuffs that are locally produced, providing the possibility to reduce the environmental footprint associated with imported feedstuffs [24]. In countries such as Greece or other Mediterranean ones, farmers’ anticipated profits can be suppressed when animals suffer from extended periods of high environmental temperatures, relative humidity and solar radiation [25]. The inclusion of soybean meal into livestock nutrition programs may also impair the profitability of the milk production, due to the high fluctuation of soybean meal cost over the period of a year. In the present study, we investigated the feasibility of partial soybean meal substitution by locally produced lupins and flaxseed. To the best of our knowledge, this is the first study to describe the effects of a flaxseed and lupin mixture (FLM) as a partial soymeal replacement on production, health and fertility parameters in dairy cows.

The partial replacement of soybean meal with FLM did not affect milk production. In the published literature, there is controversy on the effects of flaxseed supplementation on milk yield of dairy cows. A considerable increase of 6.2% in milk production was reported after feeding cows with 9.2% extruded flaxseed in the diet [26], while Neveu et al. [27] and Petit et al. [28] reported a neutral effect on milk yield when cows consumed 9% (DM) extruded and 11% whole flaxseed in the diet, respectively. Additionally, the inclusion of 3% flaxseed oil in the diets of dairy cows was found to increase milk yield; nonetheless, it was suggested that the inclusion rate should not exceed 4%, as further supplementation could act as a risk factor to decrease DMI, and thus resulting in milk yield depression [29]. The reasons for these discrepancies are not clear, and would be related to the production systems, the type of seed used (extruded or whole), the inclusion rate, which varies greatly between studies, ranging from 4% to 12,7%, and the different compositions of the rations used [30]. Despite the vast information in the literature on the effects of flaxseed on dairy performance, there is limited information on the combination of flaxseed and lupins in dairy cow diets.

In our trial, FML supplementation did not impact milk protein or fat yield. It is generally believed [30,31,32] that feeding ω-3 rich diets (such as in flaxseed) and unsaturated fats (such as contained in lupins) to dairy cows causes a reduction in fat content, which is explained by the formation of trans fatty acids that do not favor fat synthesis in the mammary cells [33]. However, in other studies, no effect on milk fat content was found when cows were fed whole flaxseed or lupins [28,34] or, on the other hand, positive effects were noted [25]. Similar controversy exists in the literature regarding the results of soybean meal substitution with lupins, where inconsistent results, both neutral and negative, are reported. In studies by Mendowski et al. [35] and Robinson and McNiven et al. [36], the exclusive use of lupin seeds as a CP source resulted in similar milk yields as the control groups where cows received soybean meal. On the contrary, Joch et al. [12] reported that a 50% substitution of soybean meal with lupins resulted in a significant decrease in milk quantity, while 30% substitution did not have a negative effect on milk production. However, the formulated diets in this trial were not isonitrogenous, thus the decline in milk yield could be attributed to the lower crude protein ratio offered to the lupin fed cows. Milk composition among control and treated groups remained similar in terms of fat, protein and lactose concentrations. Previous studies reported that the use of lupins lead to decreased milk protein percentage when the diets were formulated with complete soybean meal substitution [36,37]. This was attributed to the inferior amino acid profile of lupins; when only partial substitution was performed, no significant difference was noticed [12].

Information on the effects of a combination of dietary flaxseed and lupins on reproduction performance of cows is lacking. The published scientific data on the effects of lupins on dairy cow fertility is very limited, hence the discussion on this topic will be mostly based on the existing literature on flaxseed. Treated cows had earlier ovarian resumption than controls, and treated multiparous cows conceived earlier than controls. It has been reported that, for cows supplemented with whole flaxseed, the preovulatory follicles were larger compared to those in the control cows [38], which may be related to higher concentrations of estradiol found in cows fed ω-3 fatty acids [39]. When dairy cows are fed with flaxseed from 3 weeks pp to 100 days after calving, the estrus duration, as well as the concentration and the area under curve of estradiol, are higher than in the control cows [26]. In another experiment, Petit et al. [40] reported that cows fed a diet with 17% flaxseed have higher conception rates at first pp service compared to those consuming other fat sources; the difference was considered as the result of improved energy balance for the cows fed the flaxseed diet. Flaxseed was proven to increase the size of dominant follicles and corpora lutea compared to the control group which, in turn, resulted in higher concentrations of plasma progesterone. Moreover, cows fed on diets supplemented with flaxseed exhibited pp heat earlier and bred sooner than control cows [41]. Our results on the time of pp ovarian resumption are in agreement with that of Jahani-Moghadam et al. [42], who reported that cows fed extruded flaxseed expressed estrus earlier and had a reduced incidence of cystic ovarian disease. However, contrary to our findings, these authors reported no differences on estrus intervals and pregnancy rates. Postpartum ovarian resumption requires unhampered functionality of the hypothalamic–pituitary–ovarian (HPO) axis, which is greatly regulated by metabolic status and energy balance [43]. Delayed pp ovulation is significantly associated with a low pregnancy rate and longer calving to pregnancy intervals [44]. This is strongly related to the restoration of HPO axis functionality. Energy availability has a crucial role in controlling gonadotropins secretion [45]. In our study, it is unlikely for the observed difference in ovarian resumption to be attributed to serious metabolic status alterations, as no differences were recorded in BCS and in milk yield. Nonetheless, the NEFA concentrations, which is a very reliable index to assess adipose fat mobilization [46], differed between groups, being higher in the control animals. This might be considered as contributing factor for delayed pp ovulation in the control groups.

The results on progesterone secretion are similar with these of previous studies showing neutral effects of inclusion of either flaxseed (17% DM) [40] or lupins (8.7% DM [47], 500 g/day in ewes [48]) on progesterone synthesis.

A greater number of treated animals, compared to controls, conceived until day 120 pp, showing that the inclusion of the lupins–flaxseed mixture had a cumulative positive effect on fertility. Although these results cannot be readily explained, our findings are in direct or indirect agreement with previously published studies. For example, Petit and Benchaar [49] reported a 14.3% higher conception rate at the first pp insemination in cows fed whole flaxseed compared to those fed soybean meal. According to Petit and Twagiramungu [50], no early embryo deaths occurred when whole flaxseed was fed to cows, while the embryo mortality rates in cows that received soybean meal and a rumen protected fat (Megalac) were 8% and 15%, respectively. We assume that treated animals had either better energy utilization, as expressed with the lower NEFA concentrations, or had better uterine health, as revealed by the results of acute phase proteins and cytology. The latter hypothesis is supported by earlier studies reporting a strong association between elevated concentrations of haptoglobin during the early pp period and compromised fertility in dairy cows [51,52].

No difference was detected in the general morbidity between treated and control groups of either multiparous or primiparous cows. According to our standard protocols, animals with uncertain diagnosis for mild endometritis (score 1 at Metricheck) were reported as disease cases, albeit no treatment was undertaken. At parturition, approximately 95% of the cows are exposed to bacterial contamination. During the first two weeks pp, more than 40% of cows have some degree of uterine infection; of these animals, 30 to 35% develop subclinical endometritis which lasts until the 9th week pp [53]. It is well documented that, around parturition, tissue remodeling occurs in the myometrium and endometrium, which, along with the biochemical and endocrine changes, allow the strong uterine contractility at parturition [54]. These changes are accompanied by a significant increase in metabolic demands due to commencement of lactation; all these physiological alterations constitute a stressful condition for the cow. Among the physiological changes occurring around parturition is the increased production of acute phase proteins [55]. Acute phase proteins (APP) refer to a group of glycoproteins of hepatic origin that are synthesized in response to stimulation by specific pro-inflammatory cytokines, as well as in response to inflammation, surgical trauma, and stress [56]. Disturbances in general homeostasis, as well as metabolic changes, have been shown to lead to specific reactions, including changes in APP levels. Increased NEFA concentrations have been positively associated with elevated concentrations of APPs [55,57]. The lower cut-off points for the predictive values of APPs have not been defined. It is, however, worth mentioning that, in an effort to avoid confounding factors originating from the nature, type and severity of a disease, the assessments of NEFA and APPs levels in the current experiment were carried out only in healthy animals. Although the NEFA concentrations in both groups were below the threshold levels that are indicative for ketosis and/or excessive fat mobilization [58], the treated animals had consistently lower NEFA and APPs concentrations than the controls, which at particular time points differed significantly. It is therefore reasonable to assume that the higher NEFA concentrations induced the increased APPs levels. Alternatively, the possibility of mild subclinical disturbances that could not be diagnosed, but which might have occurred at a higher rate in the control group, should not be ruled out. 

To avoid large variation on PMN counts, we decided to examine only animals that were suspicious for mild uterine infection (cloudy mucus without pus contamination as indicated by Metricheck). The time of sampling, and the nature of the disease (Metricheck score 2), implied that treatment was not required [59]. The PMNs counts measured in both groups were indicative of mild endometritis, which confirmed the Metricheck assessment. In fact, the cutoff values of PMNs concentration to confirm subclinical (asymptomatic) endometritis vary between studies; in some studies, the threshold is set at levels as low as 8% [60], and in others as high as 18% [61]. In the present study, a remarkable evenness was observed on day 21 regarding the PMN proportion among all groups; however, on day 42 the PMN cells were fewer in treated animals. The PMN cells migrate from capillaries into the inflammation cite to kill or phagocytize bacteria [62]. Both flaxseed and lupins are rich sources of polyunsaturated fatty acids [6,63]. In humans, consumption of food rich in PUFA results in significant immune system upregulation, which is manifested by decreased proliferation of peripheral blood mononuclear cells (PBMC) in response to mitogenic stimulation, and decreased synthesis of proinflammatory cytokines such as IL-1β, IL-6 and tumor necrosis factor (TNF)-α [64,65]. It is also believed that dietary PUFA alter the composition and function of the immune system cells modulating lymphocyte proliferation, natural killer cell activity, macrophage functions and the synthesis of inflammatory cytokines produced by the macrophages [66]. When dairy cows were fed flaxseed, which is rich in ω-3 PUFAs, at 13% of DM, suppressed mitogen-driven PBMC and reduced pp blood concentration of prostaglandin E2 were observed [67,68]. As no differences were present in antigen-specific antibody responses in cattle fed flaxseed compared to cattle that received other types of fat (full fat soybean), it has been postulated that the immunomodulatory properties of the ω-3 PUFA in cattle are directed mainly through T-cell and monocyte/macrophage function [67]. Under stress conditions, incorporation of flaxseed in dairy cow diets enhances immune functions, as manifested by improved reactivity to phytohemagglutinin (PHA), anti-chicken egg albumin IgG levels and a reduction of IL-10 secretion [69]. During ketosis, when lipid mobilization occurs, the high NEFA levels are associated with reduced PMN phagocytotic ability and chemotaxis increasing the incidence of endometritis [70,71]. In the present study, the reduction rate of PMN between days 21 and 42 in multiparous and primiparous cows fed the FLM was greater than that of the controls. Throughout the sampling period, NEFA, Hpt and SAA concentrations were in general reduced in the animals fed the FLM; hence, we hypothesize that the higher NEFA levels, indicative for fat mobilization, and possibly comparatively impaired liver function, triggered a partial suppression of immune functions. Other researchers proposed that, in some instances, the presence of an increased PMN population should be considered as an indication of enhanced cellular migration, and not as an inflammation parameter [72,73]. This is unlikely to apply for our results, because both APPs and NEFA concentrations, and mainly the fertility parameters, indicated better uterine ‘health’ status in the group T animals. The differences reported in PMN clearing rate between treated multiparous cows and primiparous cows cannot be readily explained. This could possibly be attributed to the stress of calving that is much higher in heifers than in multiparous animals.

An interesting finding in our study was that oxidation markers in serum, such as TBARS and catalase activity, were similar in cows among the treated and control groups. Unsaturated fatty acids are unstable, and receptive to oxidative processes. In a study of Santos et al. [74] it was proven that flaxseed oil supplementation leads to elevated concentrations of oxidation markers in the blood of cows. Formulating rations rich in polyunsaturated fatty acids may result in the production of milk characterized by improved fatty acid profile [75], yet the risk of increased blood lipid oxidation may emerge. As fresh milk can be consumed within 5 market days, measurement of milk oxidation was assessed on two different timepoints in that interval, to better evaluate the oxidative phenomena. Results emerging from the present trial show that, when supplementing flaxseed and lupins at the levels of 1.2 kg per animal, the oxidation process in the blood did not increase.

## 5. Conclusions

The present results imply that a mixture of flaxseed and lupins can be used as a safe and profitable partial substitute for soybean meal. This conclusion is based on the neutral effect that was observed regarding milk production and milk quality characteristics, associated with the lower cost of the FML compared to soybean meal. This diet may offer an advantage to reduce the environmental burden, without negative effects on the oxidative status of animals. In addition, flaxseed and lupin seed supplementation to dairy cows may contribute to improve ovarian resumption and overall fertility during the first 120 days pp by enhancing the pregnancy rate.

## Figures and Tables

**Figure 1 animals-13-01972-f001:**
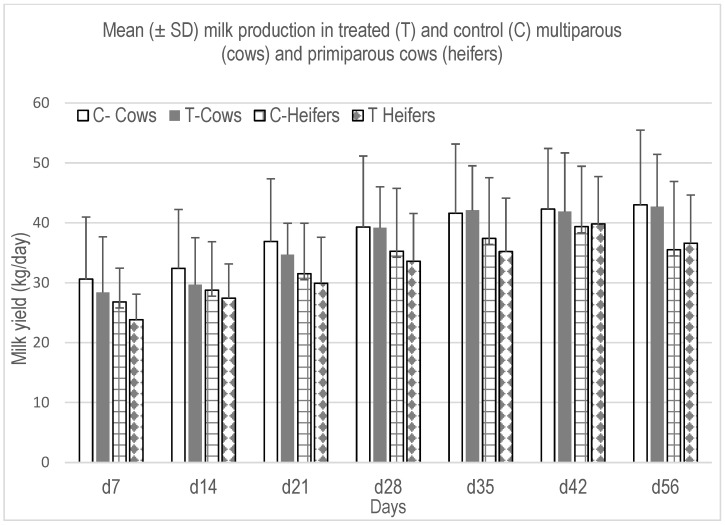
Mean milk yield during the first 8 weeks postpartum (Least Square means ± SD).

**Figure 2 animals-13-01972-f002:**
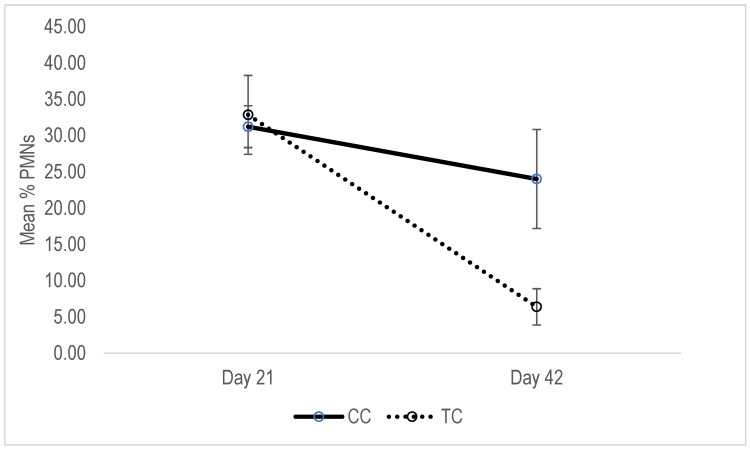
Mean of uterine epithelial PMNs concentration of multiparous control (CC) and treated cows (TC).

**Figure 3 animals-13-01972-f003:**
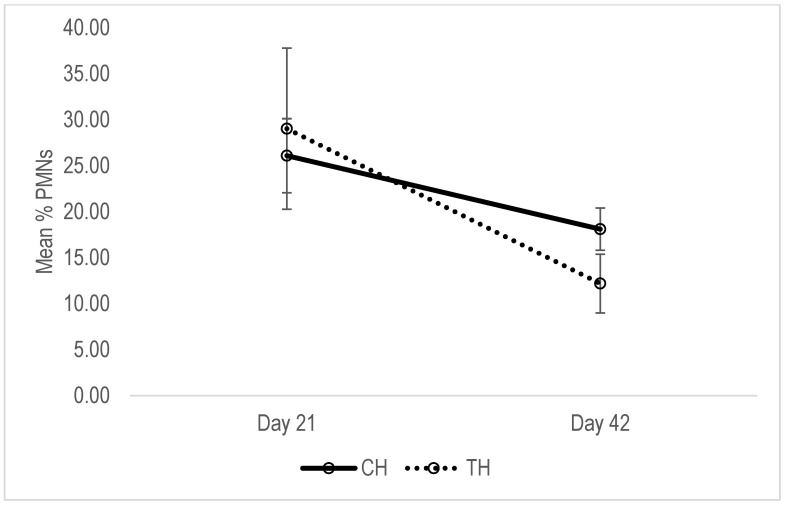
Mean of uterine epithelial PMNs concentration of primiparous control (CH) and treated cows (TH).

**Table 1 animals-13-01972-t001:** Ingredient composition and chemical composition of the TMR offered to dairy cows during the experimental period.

	Diet	
Item	Group C	Group T
Corn silage (kg/DM)	6.90	6.90
Alfalfa hay (kg/DM)	2.43	2.43
Wheat straw (kg/DM)	2.00	2.00
Corn (kg/DM)	2.61	1.74
Barley (kg/DM)	1.05	0.87
Soymeal (CP: 47%) (kg/DM)	2.46	1.23
Rapeseed meal (CP: 32%) (kg/DM)	1.85	1.76
Molasses (kg/DM)	1.93	1.54
White Lupin (kg/DM)	-	1.08
Flaxseed (kg/DM)	-	1.08
Mineral and vitamin premix ^1^ (kg/DM)	0.65	0.65
Chemical analysis ^2^		
DM (g)	21.88	21.28
CP% DM	15.85	15.91
ADF % DM	18.52	22.20
NDF % DM	37.69	41.25
Ether extract % DM	7.55	10.04
UFL/kg	0.501	0.499
Starch % DM	17.00	14.50
Sugars % DM	8.90	7.90
PDIA g/kg DM	59.45	82.31
PDI g/kg DM	54.08	76.90
RPB	43.88	45.98
Ca % DM	0.33	0.31
P % DM	0.33	0.3
Ca,absorb(g)	28.1	27.4
P,absorb(g)	24.7	25.1
Mg % DM	0.19	0.18
Na % DM	0.10	0.09
S % DM	0.28	0.25

^1^ Mineral and vitamin premix: 9.2% Ca; 4.79% P; 4.78% Mg; 1.52% S; 13.72% Na; 1.37% K; 19.5 mg/kg of Se; 23 mg/kg of I; 2.013 mg/kg of Fe; 1.068 mg/kg of Cu; 1.796 mg/kg of Mn; 2.657 mg/kg of Zn; 57 mg/kg of Co; 265 mg/kg of F; 442.000 UI/kg of vitamin A; 56.670 UI/kg of vitamin of D; and 2.630 UI/kg of vitamin E. ^2^ DM: dry matter; CP: crude protein; ADF: acid detergent fiber; NDF: neutral detergent fiber; UFL: net energy for lactation; PDIA: truly digestible dietary protein; PDI: truly digestible dietary and microbial protein; RPB: rumen protein balance.

**Table 2 animals-13-01972-t002:** Mean dry matter intake per day the first 56 days of cows fed the experimental diets ^1^.

Item	Group C ^1^	Group T	SEM ^2^	*p*-Value
Dry matter intake/day				
Multiparous cows	21.30	21.25	0.085	NS ^3^
Primiparous cows	19.40	19.45	0.078	NS

^1^ Group C: control diet, group T: diet with 50% soymeal substitution. ^2^ SEM: standard error of mean. ^3^ NS: not significant.

**Table 3 animals-13-01972-t003:** Milk composition of cows fed the experimental diets ^1^.

Item	Group C ^1^	Group T	SEM ^2^	*p*-Value
Protein (%)	3.37	3.39	0.170	NS ^3^
Fat (%)	3.95	3.88	0.325	NS
SCC (×1000)	158.5	157.0	0.086	NS
CFU (×1000)	8.20	8.75	0.158	NS

^1^ Group C: control diet, group T: diet with 50% soymeal substitution. ^2^ SEM: standard error of mean. ^3^ NS: Not significant.

**Table 4 animals-13-01972-t004:** Mean calving to first estrus intervals (only animals that expressed spontaneous estrus were included) of cows fed the experimental diets ^1^.

Type of Animals	Group C	Group T	SEM ^2^	*p*-Value
All animals	61.3	51.7	0.153	NS ^3^
Multiparous cows	60.1 ^a^	47.1 ^b^	0.128	*
Primiparous cows	62.7	56.3	0.163	NS

^1^ Group C: control diet, group T: diet with 50% soymeal substitution. ^2^ SEM: standard error of mean. ^3^ NS: not significant, *: *p* ≤ 0.05. ^ab^ Values in the same line with the same superscript do not differ significantly.

**Table 5 animals-13-01972-t005:** Pregnancy rate at day 120 pp of cows fed the experimental diets ^1^.

Type of Animals	Group C	Group T	SEM ^2^	*p*-Value
All animals	61.0	69.8	0.110	NS ^3^
Multiparous cows	57.7 ^a^	70.9 ^b^	0.074	*
Primiparous cows	67.3	68	0.035	NS

^1^ Group C: control diet, group T: diet with 50% soymeal substitution. ^2^ SEM: standard error of mean. ^3^ NS: not significant, *: *p* ≤ 0.05. ^ab^ Values in the same line with the same superscript do not differ significantly.

**Table 6 animals-13-01972-t006:** Serum TBARS and catalase activity of cows fed the experimental diets ^1^.

Item	Group C	Group T	SEM ^2^	*p*-Value
Serum TBARS (ng/mL)				
Day 1	55.58	68.72	0.282	NS ^3^
Day 3	57.02	86.66	0.850	NS
Serum catalase (nmol/min/mL)	12.35	13.09	0.147	NS

^1^ Group C: control diet, group T: diet with 50% soymeal substitution. ^2^ SEM: standard error of mean. ^3^ NS: not significant.

**Table 7 animals-13-01972-t007:** Mean (± standard deviation) serum NEFA concentration, serum haptoglobin concentration (μg/mL) and serum amyloid (mg/L) of cows fed the experimental diets ^1^.

Item	CC	TC	CH	TH
Serum NEFA (mMol/L)				
Day −7	0.39 ± 0.28	0.42 ± 0.25	0.31 ± 0.18	0.34 ± 0.21
Day 0	0.44 ± 0.36	0.49 ± 0.37	0.35 ± 0.16	0.39 ± 0.22
Day 7	0.50 ± 0.34	0.30 ± 0.22	0.47 ± 0.26	0.41 ± 0.32
Day 14	0.73 ± 0.20 *	0.43 ± 0.23	0.55 ± 0.30	0.45 ± 0.29
Day 21	0.56 ± 0.48	0.41 ± 0.29	0.56 ± 0.48	0.47 ± 0.30
Day 42	0.38 ± 0.33 *	0.21 ± 0.16	0.47 ± 0.23	0.39 ± 0.27
Serum Haptoglobin (μg/mL)				
Day −7	52.7 ± 11.1 *	35.4 ± 20.3	59.7 ± 9.8 *	39.4 ± 5.7
Day 7	26.4 ± 11.9	23.3 ± 3.4	34.5 ± 8.6 *	23.3 ± 3.4
Day 21	28.6 ± 19.4	26.9 ± 3.1	28.2 ± 5.5	26.9 ± 3.1
Serum SAA (mg/L)				
Day −7	36.8 ± 7.6	22.9 ± 4.1	78.0 ± 14.6 *	28.8 ±17.7
Day 7	22.9 ± 8.9	15.2 ± 5.2	69.3 ± 18.6	63.6 ± 13.1
Day 21	50.0 ± 10.4	30.6 ± 4.8	118.2 ± 22.5	85.5 ± 23.1

^1^ CC: control diet, multiparous cows; TC: diet with 50% soymeal substitution, multiparous cows; CH: control diet, primiparous cows; TH: diet with 50% soymeal substitution, primiparous cows. * Denotes significant difference (*p* < 0.05) between treated and control groups at the particular time point.

## Data Availability

The data presented in this study are available on request from the corresponding author. The data are not publicly available due to privacy restrictions, the trial was performed on a commercial farm and ELVIZ SA provided most of the experimental feeds.

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
