# Peer review of "Feeding Flaxseed and Lupins during the Transition Period in Dairy Cows: Effects on Production Performance, Fertility and Biochemical Blood Indices"

_animals, 2023, doi:10.3390/ani13121972_

Round 1
Reviewer 1 Report
The manuscript entitled – " Feeding flaxseed and lupins during the transition period in dairy cows: effects on production performance, fertility, and biochemical blood indices " is prepared meticulously. However, some points that need to be clarified properly before consideration have been drawn to attention.
In "Simple Summary":
In Line 22: “Milk yield and composition, as well as cheese characteristics of five replicates, were examined.”
The author should explain the findings of the cheese characteristics in the result and discussion section.
In “Abstract”:
Although there are some common abbreviations, the author should elaborate on these for the first time writing in the “Abstract” section of the manuscript. i.e. Hpt, SAA, SCC, CFU, NEFA, TBARS.
In “Introduction”:
The author should address the objectives of the study evidently. If it is cost-effective (mentioned in the simple summary), a brief cost analysis of the two diets needs to be added in the introduction section. Although feed ingredient costs differed from region to region, a cost analysis of the locally available flaxseed and lupin seed along with soybean meal should be discussed in the introduction section with the cost of the ingredients.
Another concern is that soybean meal is a protein source feed ingredient. However, flaxseed and lupin seed are oil seeds that were used as whole seeds without oil extraction. The oil seed is mostly used as a lipid source.
1. Why the whole seeds were used instead of the flaxseed and lupin seed meal?
2. What is the nutrient composition of the oil seeds (flaxseed & lupin seed) and soybean meal that were used in this experiment?
3. What are the energy values of the two diets?
4. The ether extract as %DM is much higher in the treatment group (10.4 vs 7.55%). How could it be made two diets as isoenergetic?
In "Materials & Method":
In Table 1: The elaboration of all abbreviations should include in the footnote of Table 1. i.e. DM, CP. ADF, NDF, EE, UFL, PDIN, PDIE, Ca, P.
As mentioned in the “Abstract” section “two rations were formulated to be isonitrogenous and isoenergetic (Line: 33-34)”, the chemical composition of the TMR should include both parameters.
Please describe the UFL, PDIN, and PDIE and their analysis methods in the “Materials and Methods” section. How these values were determined for two TMRs?
The “PDIN” & “PDIE” values have also differed for the two TMRs (Table 1). Please explain the reason for the differences and discuss the reflections of the two values in the results and findings.
In “Result” & “Discussion”:
Please provide all necessary information and discussion according to the comments and questions that arose after careful revision of the manuscript. Please add any new references (if added after revision) in the reference section.
Author Response
Dear reviewer we kindly thank you for your revision.
We tried to satisfy all comments and suggestions.

Reviewer 2 Report
It is an interesting study investigating the substitution of soymeal for flaxseed and lupins in the diets of dairy cows during the transition period. The study was carried out on a commercial farm, which has the advantage of reproducing the field applicability of such adjustments in diet. On the other hand, it limits the study conditions, which ultimately will restrict the inferences authors can reach with the study. My main concerns are related to those limitations and also to missing information in the materials and methods section. Altogether limits my review of the manuscript and understanding if the inferences and conclusions are adequate. Based on that I consider the manuscript cannot be accepted in its current format. Details can be found below:
Line 22: what do you mean by “cheese characteristics”? Have you evaluated cheese characteristics?
Line 22: what do you mean by “5 replicates”? It disagrees with the information in the main body of the manuscript.
Lines 25, 46, 388, 391, and 568: any implication on profitability is an assumption. You did not perform an economic impact evaluation.
Line 31: 85 days? 25+56 = 81 days
Line 79: substitute “amd” by and”
Lines 125-130: You described the cows were randomly distributed in 2 groups. More details especially related to housing and feeding would be very important. Without those details, readers would assume cows were allocated in groups, which would imply that pen would be your experimental unit. If so, the study would need replicated pens as replicates. Can you be more specific in this regard?
Line 130: for hematological and cytological parameters you used 60 animals,(15 Multiparous – C + 15 heifers + 15 Multiparous – T + 15 heifers – T). How were the cows allocated?
Line 132: You mentioned you substitute “50% of the soybean meal by flax and lupin seeds mixture.” It is not clear what was the basis of the substitution. Was it a mass basis? CP basis? MP basis? The way it is written it gives me the impression that it was the mass basis, however, it does not match the information in table 1.
Line 141 and Table 1: For the ingredient formulation of the diet use kg/kgDM instead of kg per cow. It will make it difficult for readers to reproduce this diet later on or for scientists to use this information in the future.
A study of this nature would require more complete diet information. For appropriate understanding of the study and results, readers would need information of MP, RDP, RUP, ME, amino acids (if it is possible), and FA profile.
You mention that 50% of the soymeal was substitute by UFL is limited information regarding the energy of the diet. You would prefer to show metabolizable energy instead.
Line 153: do you mean 330 cows?
Line 161: do you mean 330 cows?
Line 209: do you mean 330 cows?
Lines 261-275: there is missing information regarding the statistical analysis. Which experimental design have you adopted? What was your experimental unit? Which effects have you considered fixed and random? Which variance-covariance matrix have you considered for analyzing the repeated measures? How have you considered the effect of lactation order?
As the above information is missing or omitted it became harder to review the results, discussion, and conclusion sections
Author Response

(The authors gave the same response as above.)

Reviewer 3 Report
Feeding flaxseed and lupins during the transition period in dairy cows: effects on production performance, fertility and bio-chemical blood indices
Ioannis Nanas, Stella Dokou, Labrini Athanasiou, Eleni Dovolou, Thomas Chouzouris, Stelios Vasilopoulos, Katerina Grigoriadou, Ilias Giannenas and Georgios S. Amiridis.
General comment.
The authors invest a lot of time and effort to run the experiment, taking a lot of samples. However, the study has some main flaws.
Specific comments.
Abbreviations like SCC, UFC and UFL should be written in full the first time, and after that only abbreviations. Do the same for all abbreviations throughout the whole document.
Wat it is UFL. The international measure of energy is Joule, what is the justification to use a different one?
There is a misspelling at the beginning of line 79. “amd”, please correct it.
In table 1 ingredients are given in kg/cow. However, it differs from the title. The correct thing to do is to report the percentages of each ingredient in the diet expressed as % of dry matter in the diet. So the sum of all ingredients adds up 100%.
It is confusing to present diet´s 60% of DM and, DM kg 23.2 What does this last value represents? The kg of DM intake per cow? The right thing to do is to report just the diet´s DM specifying “as fed” (i.e. “DM, % as fed”), deleting DM kg to avoid confusion.
What does PDIN and PDIE stand for?
It is not reported how the chemical composition of diet´s ingredients and TMR were assessed.
Although NIRS it is a very useful tool to asses the chemical composition of feeds, for experimental purposes it is recommended to asses (corroborate) the nutritional composition of ingredients ad diets using wet chemistry since there are discrepancies between the two methods.
Do the authors used a program to balance the diet? If so, please report it.
Does the milk components were assessed only during the morning milking? If so, it is a mistake. Milk composition must be assessed during two consecutive days (four milkings).
Authors must indicate the experimental design of the experiment according to the variables analysed, and include the equations.
Authors must start the results section presenting the results of the nutritional composition of the diets by treatment and by experimental period (week).
Body weight was not assessed. Why? It is a trait as important as BSC. You cannot assessed the effect of a diets on BCS without assessing bodyweight.
After the presentations of the nutritional composition of the diets, authors must present the effect of diets on animal performance variables starting with dry matter intake (kg/cow/day), milk yield (kg/cow/day), milk composition (fat, protein and lactose g/kg and kg/day) etc. Than, by experimental period (week).
It is a mistake to present result in Figures and in tables. Authors must choose only one way to present the results. I recommend to present animal performance variables in tables rather the in figures.
Table 1 must include all animal performance variables (i.e. BCS etc.).
Line 344. Replace “first calf heifers”, with first parity cows. Correct the rest of the document accordingly.
Eighty-one references seems to me excesived.
Author Response

(The authors gave the same response as above.)

Reviewer 4 Report
Manuscript animals-2251636, entitled “Feeding flaxseed and lupins during the transition period in dairy cows: effects on production performance, fertility and biochemical blood indices”
Recommendation: The above paper is not suitable for publication in its present form.
The article provides useful information about the effects of feeding flaxseed and lupins on production performance, fertility and biochemical blood indices during the transition period in dairy cows. Although, the experiment was in general appropriately designed and implemented, there are some points that should be corrected or clarified.
General comments
· Please clarify sampling days and sample size for TBARS and catalase.
· Where are the data for serum progesterone and PAG presented?
· The section 2.1 is not necessary. Please add it at the end before the references
· In Table 3, please provide superscripts indicating significant differences among time points within a column
· In some cases, data presented in Table 3 are different from that discussed in text. For example, in text significant differences are presented, while P-values in Table are NS. Please check L297-298, 310, 327 etc
· In some cases, SEM is greater than means, i.e. NEFA – Multiparous – Day 42. How was the difference significant? On the other hand, for serum SAA – Multiparus and Primiparous, SEM was too small. How were these differences not significant?
Minor points
L22: What do you mean by “of five replicates”
L23: “dietary treatment” instead of “feed modification”
L30: “selected” instead of “used”
L35: “…composition, SCC content and CFU on a weekly basis. Blood samples were also weekly collected and serum…”
L40: Cows or multiparous cows?
L51: “due to its” instead of “because of the”
L61: Please rephrase “and overall farm concurrently lower environmental cost”
L88: “scarce” instead of “limiting”. Please add references
L90: “along” instead of “simultaneous”
L97: Please delete “(Drackley and Cardoso, 2014)”
L98: “candidates” instead of “alternatives”
L104: “in” instead of “of”
L128-130: Multiparous or primiparous?
L201-203: Please rephrase
L251-252: Please rephrase
L273: “performed” instead of “done”
L274: “carried out” instead of “run”
L295: “Table 3”
L298-299: “…effect on Hpt concentration on days 7 and 21 (P = 0.248).”
L306: “…cows are also presented…”
L330: “…no significant differences were observed between…”
L339: “dietary treatment” instead of “feed modification”
L345: “presented” instead of “given”
L384: “Diet” instead of “Feed”
L400: “…the diet [27], while Neveu et al. [28] and…”
L421: “as” instead of “than”
L422: “On the contrary” instead of “Contrariwise”
L424: “quantity” instead of “output” and “However” instead of “Overall”
L427: “similar” instead of “indistinguishable”
L434: “Information on the effects of a combination…”
L444: “In another experiment, Petit et al. [41]…”
L449: Please add reference
L451-452: “…resumption are in agreement with that of Jahani-Moghadam et al. [43], who…”
L454: “on days open”?
L458: “risk” or “possibility”?
L466: “…are similar with these of previous studies…”
L467-468: “of either flaxseed (17% DM) [24] or lupins (8.7% DM [50], 500g/day in ewes [51]) on…”
L470: “A greater number of” instead of “More”
L508-509: NS differences in most cases
L564: “present”
L565: “…kg per animal serum oxidation process was not accelerated.”
Author Response

(The authors gave the same response as above.)

Round 2
Reviewer 1 Report
Thank you for your thorough revision.
Author Response
We thank reviewer for his comments and support to improve our work.
Reviewer 2 Report
I still think it is an interesting study investigating the substitution of soymeal for flaxseed and lupins in the diets of dairy cows during the transition period. And I still have the major concerns raised in the previous review. The authors added some missing information, but some others are still missing,
The cows were randomly distributed in 2 groups. More details especially related to housing and feeding is very important.Such information is the one that would allow reviewers to understand the appropriateness of the design, the number of replicates, etc. In the response letter, the authors responded the cows had ear tag identification. It does not bring information on intake measurement for instance.
Based on the provided pieces of information in the Materials and methods I’ll assume cows were pen housed, and that all cows in group C were in 1 pen and all cows in group T were in another pen. Is it correct? If so, the study had only one replicate for treatment. For more details in this regard, I suggest the paper of St-Pierre, 2007.
St-Pierre NR. Design and analysis of pen studies in the animal sciences. J Dairy Sci. 2007 Jun;90 Suppl 1:E87-99. doi: 10.3168/jds.2006-612.
Once again, the missing information limits my review of the manuscript, and my understanding if the inferences and conclusions are adequate. Based on that I consider the manuscript cannot be accepted in its current format.
Author Response
We thank reviewer for his valuable comments and reply in detail in the attachement.

Reviewer 3 Report
I´m satisfied with the current state of the manuscript.
Author Response
We thank reviewer for his valuable comments and we reply in detail.

Reviewer 4 Report
Authors made the majority of the necessary amendments. However, some points should be corrected before the acceptance of their article.
Sampling days and size for TBARS and Catalase should be added. What do you mean by 1 and 3 days after blood collection in Table 4?
L134: "orts"?
L330: "...no significant differences were..."
L452: Please add reference
L564-566: Please rephrase
Author Response
We thank reviewer for his valuable comments and help to improve the presentation of our work.